# Characterization of *Triadica sebifera* (L.) Small Extracts, Antifeedant Activities of Extracts, Fractions, Seed Oil and Isolated Compounds against *Plutella xylostella* (L.) and Their Effect on Detoxification Enzymes

**DOI:** 10.3390/molecules27196239

**Published:** 2022-09-22

**Authors:** Shudh Kirti Dolma, S. G. Eswara Reddy

**Affiliations:** 1Entomology Laboratory, Agrotechnology Division, CSIR-Institute of Himalayan Bioresource Technology, Palampur 176061, India; 2Academy of Scientific and Innovative Research (AcSIR), Ghaziabad 201002, India

**Keywords:** botanicals, feeding deterrence, repellence, synergistic, GST, AChE

## Abstract

*Plutella xylostella* L. is one of the world’s major pests of cruciferous crops. The indiscriminate use of synthetic insecticides has led to insecticide resistance and resurgence, and has been harmful to non-target organisms and the environment. Botanical insecticides are the best alternatives to synthetic pesticides for the management of pests in organic agriculture and integrated management. *T. sebifera* is an invasive species and has good potential as an insecticide due to the availability of plant material in some parts of India. The antifeedant activities of *T. sebifera* have not been reported against *P. xylostella* and other lepidopteron insects to date. Therefore, the current study targeted the characterization of leaf and bark extracts, feeding deterrence, synergistic and detoxification enzyme activities of leaf/bark ethanolic extracts/fractions, seed oil, and isolated compounds. UHPLC-QTOF-IMS analysis showed that shikimic acid, xanthoxylin, quercetin, kaempferol, methyl gallate, and stigmasterol are common metabolites identified in leaf and bark extracts. The combination of seed oil with bark extract showed higher deterrence (DC_50_ = 317.10 mg/L) as compared to leaf/bark extracts alone. Gallic acid showed higher deterrence (67.48%) than kaempferol and quercetin. The *n*-butanol fraction of bark was more repellent (RC_50_ = 414.61 mg/L). Based on DC_50_, the seed oil with leaf extract (1:1 ratio) alone with choice and seed oil with leaf and bark extract without choice showed synergistic interaction, but seed oil with bark extract with choice showed additive interaction. The ethanol extract of leaf, bark, and seed oil inhibited GST and AChE in *P. xylostella.* The leaf extract and seed oil or their combinations may be recommended as antifeedants to reduce damage by *P. xylostella* based on persistence, antifeedant, phytotoxicity, safety to predators/parasitoids, etc., under field conditions.

## 1. Introduction

Diamondback moth, *Plutella xylostella* L., is one of the most serious pests of cruciferous crops in India and the world that cause economic damage [1,2]. Synthetic insecticides are widely used for the control of *P. xylostella* all year round. The indiscriminate and repeated use of the same group of insecticides has led to insect resistance [3,4], insecticide residues, environmental contamination, harm to consumer’s health, and natural enemies of pests [5]. In order to decrease the use of synthetic pesticides, plant-based botanical insecticides have been suggested as a top priority for insect control due to environmental safety and non-target organisms. At present more importance is given to botanical and other biopesticides for the control of pests due to resistance, safety, and environmental issues.

Around the world, and particularly in India, limited botanical formulations are commercially available for the control of insect and mite pests. Therefore, it is necessary to screen and identify the lead(s) from plants for the development of botanical formulations against the target pest(s).

*Triadica sebifera* (L.) Small (Euphorbiaceae), is an invasive species native to China and has been introduced to tropical, subtropical, and temperate regions of the globe. It has been distributed to Asia, Africa, the USA, the West Indies, Australia, French Polynesia and Hawaii, the Soviet Union, and the Black Sea in Georgia. *T. sebifera* has been used as an anti-bacterial, anti-microbial, anti-viral, antioxidant, and anti-inflammatory [6,7,8,9]. Seeds are used for herbal medicines, cosmetics, and pharmaceuticals [10,11], whereas root and bark extracts are for snake bites and skin ulcers [12]. In India, *T. sebifera* was introduced to Saharanpur, Uttar Pradesh, in 1858 [13], and then transported to Dehradun (Uttarakhand, India). In addition to the Himalayan states [14,15,16], it has also been found in the eastern and southern states of India [17,18].

*T. sebifera* was selected in the present study due to the availability of plant material and seeds in different parts of India (Himachal Pradesh, Uttarakhand, Jammu and Kashmir, Uttar Pradesh, and Tamil Nadu) to study the insecticidal activities against *P. xylostella*. No report on insecticidal activities of *T. sebifera* against *P. xylostella* has been reported to date except for *Aphis craccivora* [19]. Therefore, in the present study, we explored the characterization, antifeedant/feeding deterrence, repellent, synergistic, and detoxifying enzyme inhibition activities of seed oil, leaf, bark ethanol aqueous extract, and its fractions against larvae of *P. xylostella*.

## 2. Results

### 2.1. Identification and Characterization of Metabolites in Leaf and Bark Ethanol Aqueous Extract of T. sebifera

A total of sixteen peaks have been identified in leaf and bark ethanol aqueous extracts of *T. sebifera,* while one major peak present in both samples remains unidentified. The metabolites have been annotated based on RT, comparison of molecular weight (*m*/*z* ratio), and mass fragmentation pattern in the literature (Table 1; Figure 1). Shikimic acid (**1**), xanthoxylin (**2**), quercetin (**3**)**,** kaempferol (**4**)**,** methyl gallate (**5**)**,** and stigmasterol (**12**) were some common metabolites annotated in both the extracts. Apart from these, scopoletin (**9**), β-sitosterol (**11**), stigmasterol glycoside (**13**)**,** kaempferitrin (**16**)**,** along with one coumarin derivative type compound (**6**) and one glycosidic compound (**7**) were annotated from bark ethanol aqueous extract. In contrast, cinnamic acid (**8**), gallic acid (**10**), astragalin (**14**), and isoquercetin (**15**) were annotated in the leaf ethanol aqueous extract of *T. sebifera* [20,21,22,23,24,25].

Shikimic acid (**1**) and xanthoxylin (**2**) were annotated in the same peak at RT 4.155 min with a protonated ion peak at *m*/*z* 175.15 (M+H)^+^ and *m*/*z* 197.13 (M+H)^+^, respectively. The shikimic acid was annotated from its MS/MS fragments at *m*/*z* 175 (M+H)^+^, 174 [C_7_H_10_O_5_ (M)]^+^, and 130 [C_6_H_10_O_3_ (M-COOH)]^+^, while fragments for xanthoxylin were annotated at *m*/*z* 197 (M+H)^+^, 196 [C_10_H_12_O_4_ (M)]^+^, and 180 [C_10_H_12_O_3_ (M-OH)]^+^. Similarly, quercetin (**3**), kaempferol (**4**), and methyl gallate (**5**) were annotated at RT 4.705, 4.889, and 5.739 min in both the ethanol aqueous extracts with *m*/*z* 303.10 (M+H)^+^, 287.11 (M+H)^+^ and 207.02 (M+Na)^+^, and 185.04 (M+H)^+^, respectively. Quercetin (**3**) was further annotated from its mass fragments observed at *m*/*z* 303 (M+H)^+^ and 197 [C_9_H_8_O_5_ (M+H-C_6_H_6_O_2_)]^+^. The mass fragment at *m*/*z* 197 was observed after losing the C_6_H_6_O_2_ unit from the main moiety, confirming its quercetin identity (**3**). The MS/MS fragments for kaempferol (**4**) were annotated at *m*/*z* 287 (M+H)^+^ and 182 [C_9_H_8_O_4_ (M+2H-C_6_H_6_O_2_)]^+^. Similarly, methyl gallate (**5**) was further annotated from its sodiated and protonated ion peaks observed at *m*/*z* 207.02 (M+Na)^+^ 185.04 (M+H)^+^, respectively. One coumarin derivative type compound (**6**) at RT 6.538 min with *m*/*z* 383.13 (M+H)^+^ and one glycosidic compound (**7**) at RT 7.455 min having a protonated ion peak at *m*/*z* 503.16 (M+H)^+^ was annotated in the bark ethanol aqueous extract of *T. sebifera.* The coumarin derivative compound (**6**) was annotated from its mass fragment peak observed at *m*/*z* 383.13 (M+H) ^+^ and 163 (C_9_H_6_O_3_+H)^+^, which was the basic moiety of coumarins (7-hydroxy coumarin or umbelliferone). Previously some coumarins and their derivative type compounds were reported from *T. sebifera* [21]. Similarly, the fragments for glycosidic compound (**7**) were annotated at *m*/*z* 503 (M+H)^+^ and 341 (M+H-glu)^+^, which were observed after the loss of glucose moiety. The above two fragments indicate the loss of glucose moiety from the main unit and confirm it as a glycosidic molecule. Cinnamic acid (**8**)**,** scopoletin (**9**)**,** and gallic acid (**10**) were annotated at RT 7.557, 7.639, and 7.682 min with protonated ion peaks at *m*/*z* 149.11 (M+H)^+^, 193.05 (M+H)^+^ and 341.11 (2M+H)^+^, and 171.09 (M+H)^+^, respectively. The cinnamic acid (**8**) was further annotated from its mass fragments observed at *m*/*z* 149 (M+H) ^+^ and 148 (M)^+^. Similarly, scopoletin (**9**) was also annotated from its mass fragments observed at *m*/*z* 193 (M+H)^+^ and 163 [C_9_H_6_O_3_ (M+H-OCH_3_)]^+^. The mass fragments observed for gallic acid (**10**) were 171.09 (M+H)^+^ and 127 [C_6_H_6_O_3_ (M+H-COOH)]^+^. The β-sitosterol (**11**) and stigmasterol (**12**) were two phytosterols annotated in the extracts at RT 8.187 and 8.367 min with *m*/*z* 415.15 (M+H)^+^ and 413.20 (M+H)^+^. Along with these, one stigmasterol glycoside (**13**) was also annotated in bark *ethanol aqueous* extract at RT 12.647 min with *m*/*z* 575.20 (M+H)^+^, which was annotated from their MS/MS fragments at 575 (M+H)^+^ and 413 [C_29_H_48_O (M+H-glu)]^+^. The mass fragment at *m*/*z* 413 [C_29_H_48_O (M+H-glu)]^+^ was observed after loss of glucose moiety, and *m*/*z* 413 was annotated as the protonated mass for the stigmasterol (C_29_H_48_O) skeleton. Similarly, three flavonoid glycosides as astragalin (**14**)**,** isoquercetin (**15**), and kaempferitrin (**16**) were annotated at RT 12.759, 13.187, and 15.474 min having *m*/*z* 449.17 (M+H)^+^, 487.07 (M+Na)^+^/465.09 (M+H)^+^, and 579 (M+H)^+^, respectively. Astragalin (**14**) was annotated by its main protonated ion peak observed at *m*/*z* 449 (M+H)^+^ and its main fragment was observed at 287 [C_15_H_10_O_6_ (M+H-glu)]^+^, indicating its basic skeleton as kaempferol, which was observed after the loss of glucose moiety. However for isoquercetin (**15**), the sodiated and protonated ion peaks were observed at *m*/*z* 487 (M+Na)^+^ and 465 (M+H)^+^. The major observed MS/MS fragment at 303 [C_15_H_10_O_7_ (M+H-glu)]^+^ confirmed the presence of glucose moiety in the skeleton. Similarly, for kaempferitrin (**16**), the mass fragments were observed at *m*/*z* 433 [C_21_H_20_O_10_ (M+H-rha)]^+^ and 287 [C_15_H_10_O_6_ (M+H-rha-rha)]^+^, which were observed after the simultaneous loss of two rhamnose units from the structure. The basic moiety for astragalin (**14**) and, kaempferitrin (**16**) was kaempferol annotated from their *m*/*z* 287 (M+H)^+^. Apart from these, one major peak (**17**) was observed in both the samples at RT 18.794 min, which remains unidentified.

### 2.2. Antifeedant/Feeding Deterrent Activity of Leaf, Bark Extracts, Seed Oil, Isolated Compounds, Binary Mixtures, and Fractions against P. xylostella

The antifeedant/feeding deterrence of ethanol aqueous leaf, bark extract, seed oil, binary mixtures, fractions, and isolated compounds with and without the choice method against *P. xylostella* in terms of deterrent concentration (DC_50_) and percent feeding deterrence index (FDI) are presented in Table 2, Table 3 and Table 4.

#### 2.2.1. Leaf, Bark Extracts, and Seed Oil

With and without the choice method, in terms of DC_50_, bark ethanol aqueous extract showed promising feeding deterrence activity (DC_50_ = 2420.83 and 3678.02 mg/L, respectively) against *P. xylostella* and was followed by leaf ethanol aqueous extract (DC_50_ = 3624.80 and 3944.50 mg/L) and seed oil (DC_50_ = 9079.59 and 4019.85 mg/L), respectively (Table 2).

Among extracts and seed oil, with and without the choice method, the feeding deterrence of the leaf and bark ethanol aqueous extract at 1% was significantly higher (68.94–77.14%) as compared to other concentrations (Appendix A). Less deterrence was observed (6.05–21.62%) at a lower concentration. Similarly, the feeding deterrence was higher at a higher concentration of seed oil (54.15–67.55%) and lowest at a lower concentration (6.05–17.37%). The leaf, bark ethanol aqueous extract, and seed oil of *T. sebiferum* were not superior to the positive control, i.e., Indo-Neem (azadirachtin 0.15% EC) at 5 mL L^–1^ with and without the choice method (DC_50_ = 2024.58 and 2873.99 mg/L, respectively) except bark ethanol aqueous extract with choice.

#### 2.2.2. Binary Mixtures of Seed Oil with Leaf and Bark Extracts

The feeding deterrent activity of combination/binary mixtures of seed oil with leaf and bark ethanol aqueous extract (1:1 ratio) against *P. xylostella* in terms of deterrent concentration (DC_50_), percent FDI with and without the choice method is presented in Table 2. With the choice method, the combination of seed oil with bark ethanol aqueous extract reported more promising deterrence (DC_50_ = 1053.05 mg/L) as compared to seed oil with leaf ethanol aqueous extract (DC_50_ = 1328.66 mg/L). Similarly, without choice also, seed oil with bark ethanol aqueous extract was found more deterrence (DC_50_ = 317.10 mg/L) as compared to seed oil with leaf ethanol aqueous extract (DC_50_ = 383.28 mg/L). With respect to percent deterrence, among the binary mixtures, with and without the choice method, the feeding deterrence of seed oil with bark ethanol aqueous extract (1:1 ratio) at the higher concentration of 1000 mg/L was significantly (F_4,49_ = 5.42 to 24.19; *p* < 0.001) higher (51.91–77.75%) against *P. xylostella* and was followed by seed oil with leaf ethanol aqueous extract (49.57–51.91%) as compared to other concentrations. Less feeding deterrence (3.57–16.04%) was observed at the lower concentration of 62.5 mg/L (Appendix A). The binary mixtures of seed oil with leaf and bark extracts are superior and showed higher feeding deterrence as compared to Indo-Neem (azadirachtin 0.15% EC) with choice (DC_50_ = 2024.58 mg/L) and no choice (DC_50_ = 2873.99 mg/L).

#### 2.2.3. Leaf and Bark Fractions

The antifeedant/feeding deterrent activity of leaf and bark fractions of *T. sebifera* against *P. xylostella* in terms of DC_50_ and percent FDI is presented in Table 3.

##### Leaf Fractions

With the choice method, the ethyl acetate fraction was more deterrent (DC_50_ = 265.16 mg/L) to larvae of *P. xylostella* and was followed by the *n*-hexane fraction (DC_50_ = 755.51 mg/L) as compared to the *n*-butanol and water fractions (DC_50_ = 1063.14 and 1504.53 mg/L, respectively). Similarly, without the choice method, the water fraction was more deterrent (DC_50_ = 219.88 mg/L) followed by the *n*-butanol and ethyl acetate fractions (DC_50_ = 235.75 and 283.68 mg/L), respectively as compared to the *n*-hexane fraction (DC_50_ = 577.71 mg/L) (Table 3). All leaf fractions of *T. sebiferum* were superior to the positive control, i.e., Indo-Neem (azadirachtin 0.15% EC) at 5 mL L^–1^ with and without the choice method (DC_50_ = 2024.58 and 2873.99 mg/L, respectively). With respect to percent deterrence by the choice and no choice method, ethyl acetate at the higher concentration (1000 mg/L) reported significantly (F_4,49_ = 6.99 to 28.25; *p* < 0.0001) higher feeding deterrence (75.17–80.23%) against *P. xylostella* and was followed by the *n*-butanol and methanol fractions (45.10–75.37%) as compared to other concentrations. In the lower concentration at 62.5 mg/L, less deterrence was observed (6.43–20.26%) (Appendix A).

##### Bark Fractions

With the choice method, the *n*-hexane fraction was more deterrent (DC_50_ = 455.41 mg/L) against larvae of *P. xylostella* and was followed by the ethyl acetate fraction (DC_50_ = 727.68 mg/L) as compared to water and the *n*-butanol fractions (DC_50_ = 1447.12 and 2189.43 mg/L, respectively). Similarly, without the choice method, the ethyl acetate fraction was more deterrent (DC_50_ = 318.37 mg/L) followed by the *n*-butanol and water fractions (DC_50_ = 411.09 and 488.20 mg/L, respectively) as compared to the *n*-hexane fraction (DC_50_ = 573.02 mg/L) (Table 3). Among bark fractions (with and without the choice method), ethyl acetate at higher concentration (1000 mg/L) reported significantly (F_4,49_ = 4.24 to 21.73; *p* < 0.005) higher feeding deterrence (54.00–99.68%) against *P. xylostella* and was followed by the *n*-hexane fraction (63.25–96.24%) as compared to other fractions and concentrations. The lower concentration at 62.5 mg/L showed less deterrence (10.05–28.57%) (Appendix A). The bark fractions showed more feeding deterrence than the positive control, i.e., Indo-Neem (azadirachtin 0.15% EC) at 5 mL L^–1^ with and without the choice method (DC_50_ = 2024.58 and 2873.99 mg/L, respectively) except butanol with choice.

#### 2.2.4. Isolated Compounds

The isolated compounds, viz., kaempferol-3-*O*-glucoside, quercetin-3-*O*-glucoside, gallic acid, and shikimic acid were evaluated for their feeding deterrence against *P. xylostella* in comparison with the positive control (Indo-Neem). Among the compounds (with choice), gallic acid showed significantly (F_4,49_ = 8.20; *p* < 0.0001) higher feeding deterrence (67.48 ± 1.99%) and was at par with kaempferol-3-*O*-glucoside (66.15 ± 2.10%) and quercetin-3-*O*-glucoside (64.63 ± 1.96%) and was superior to the positive control (54.97 ± 2.25%) and shikimic acid (56.50 ± 1.78%). Similarly, without the choice method, gallic acid was found significantly (F_4,49_ = 3.13; *p* < 0.02) higher feeding deterrence (63.8 ± 2.39%) and was at par with Indo-Neem (58.02 ± 2.56%) and quercetin-3-*O*-glucoside (57.63 ± 1.54%) followed by kaempferol-3-*O*-glucoside (55.75 ± 1.25%) (Table 4).

### 2.3. Growth Inhibition Activity of Leaf, Bark Extracts, Its Fractions, Seed Oil, Isolated Compounds, and Binary Mixture

The growth inhibition activity of ethanol aqueous leaf, bark extract, its fractions, and the seed oil of *T. sebifera* against *P. xylostella* after 48 h of treatment is presented in Appendix A.

#### 2.3.1. Leaf, Bark Extracts, and Seed Oil

Based on IC_50_, leaf ethanol aqueous extract showed promising larval growth inhibition (IC_50_ = 2696.49 mg/L) as compared to seed oil (IC_50_ = 3225.25 mg/L) (Appendix A). However, bark ethanol aqueous extract showed homogeneity (*p* < 0.15) between the concentrations studied. With respect to the growth inhibition of larvae of *P. xylostella*, a higher concentration of bark ethanol aqueous extract at 10,000 mg/L showed a significantly higher GIR of 99.77% (F_4,49_ = 21.66; *p* < 0.0001) and was followed by leaf ethanol aqueous extract and seed oil (89.40 and 82.37%, respectively) as compared to other concentrations evaluated (Appendix A). The higher concentrations of leaf, bark extracts, and seed oil showed more growth inhibition of larvae as compared to the positive control, i.e., Indo-Neem (65.68%).

#### 2.3.2. Leaf and Bark Fractions

The growth inhibition activity of leaf and bark fractions of *T. sebifera* against *P. xylostella* after 48 h of treatment was presented in Appendix A. Based on IC_50_, the *n*-butanol leaf fraction showed promising larval growth inhibition (IC_50_ = 273.55 mg/L) as compared to the water fraction (IC_50_ = 424.55 mg/L). However, *n*-hexane and ethyl acetate leaf fractions showed homogeneity (*p* < 0.15) between the concentrations studied. With respect to percent GIR, the *n*-hexane fraction reported 100% inhibition at 1000 mg/L followed by the *n*-butanol fraction (90.86%) as compared to the ethyl acetate and water fractions (77.22 ± 4.56 and 68.65 ± 5.04, respectively) (Appendix A). The higher concentration of leaf fractions was at par with 500 mg/L. Based on IC_50_, the water fraction of bark showed higher inhibition (IC_50_ = 315.07 mg/L), but the *n*-hexane, ethyl acetate, and *n*-butanol fractions showed homogeneity (*p* < 0.15) between the concentrations studied (Appendix A). With respect to percent GIR, among bark fractions, the ethyl acetate fraction showed 99.68% inhibition at 1000 mg/L followed by the *n*-hexane fraction (96.24%) as compared to the *n*-butanol and water fractions (87.36 and 79.92%, respectively) (Appendix A). The GIR of bark fractions at higher concentrations was at par with 500 mg/L (52.3 to 86.36%). The higher concentrations of leaf/bark fractions also showed more GIR as compared to the positive control, i.e., Indo-Neem (65.68%).

#### 2.3.3. Isolated Compounds

The growth inhibition of isolated compounds against larvae of *P. xylostella* was presented in Table 4. Gallic acid showed a significantly (F_4,49_ = 14.56; *p* < 0.0001) higher GIR (59.02%) and was at par with shikimic acid (57.52%) and quercetin-3-*O*-glucoside (57.34%) followed by kaempferol-3-*O*-glucoside (45.08%). All the isolated compounds are not superior to the positive control (Indo-Neem), which showed higher GIR (65.68%).

#### 2.3.4. Binary Mixtures 

The growth inhibition activity of binary mixtures of seed oil with leaf and bark ethanol aqueous extract of *T. sebifera* without the choice method against *P. xylostella* after 48 h of treatment is presented in Appendix A. Results showed that both the blends at 1000 mg/L showed significantly more GIR (98.7 to 99.87%) and were at par with 250 and 500 mg/L (80.44 to 88.06%) as compared to the other two lower concentrations (62.5 and 125 mg/L), which showed 54.28 to 72.56%. The binary mixtures are more superior for their feeding deterrence as compared to Indo-Neem (65.68 ± 1.60%).

### 2.4. Repellent Activity of Leaf and Bark Fractions 

Among leaf fractions, the water fraction was more repellent (RC_50_ = 540.05 mg/L) against *P. xylostella* and was followed by the *n*-hexane and *n*-butanol fractions (RC_50_ = 557.49 and 565.18 mg/L, respectively) as compared to the ethyl acetate fraction (RC_50_ = 638.63 mg/L) (Table 5). Similarly, among bark fractions, the *n*-butanol fraction was more repellent (RC_50_ = 414.61 mg/L) followed by the *n*-hexane fraction (RC_50_ = 629.58 mg/L) as compared to the ethyl acetate and water fractions (RC_50_ = 773.76 and 834.48 mg/L, respectively). With respect to percent repellency in leaf fractions, the *n*-hexane and water fractions at 10,000 mg/L showed significantly (F_4,34_ = 14.81; *p* < 0.0001) higher repellence (87.14%) and were followed by the ethyl acetate and *n*-butanol fractions (80%) (Appendix A) and were at par with 5000 mg/L and were followed by 2500 mg/L (64.28 to 74.28%) as compared to lower concentrations (47.14 to 61.43% repellency). Among bark fractions, the *n*-hexane fraction at 10,000 mg/L also reported significantly higher repellence of 92.86% (F_4,34_ = 20; *p* < 0.0001) and was followed by ethyl acetate (90%) as compared to the water and *n*-butanol fractions (80.00 and 77.14%, respectively) (Appendix A). All the bark fractions at higher concentrations were at par with 5000 mg/L.

### 2.5. Joint Action Studies of Binary Mixtures of Seed Oil with Leaf and Bark Ethanol Aqueous Extract of T. sebifera against Larvae of P. xylostella

The joint action studies (synergistic, additive, and indifferent) of binary mixtures of seed oil with leaf and bark ethanol aqueous extract against larvae of *P. xylostella* are presented in Table 6. Fractional effect indices (FEI) with the choice method based DC_50_, synergistic interaction between the seed oil with leaf ethanol aqueous extract (0.513) and additive interaction (0.726) between seed oil with bark ethanol aqueous extract was observed. Similarly, without the choice method, there was more synergistic interaction between the seed oil with bark ethanol aqueous extract (0.167) as compared to seed oil with leaf ethanol aqueous extract (0.193). Similarly, based on percent FDI with and without the choice method, an indifferent type of interaction was observed between both the mixtures of seed oil with leaf and bark ethanol aqueous extract (1:1 ratio). In the case of growth inhibition rate (GIR), an indifferent type of interaction was observed between both the mixtures/blends of seed oil with leaf and bark ethanol aqueous extract of *T. sebifera*.

### 2.6. Detoxification Enzyme Inhibition Activities of Leaf, Bark, and Seed Oil of T. sebifera in P. xylostella

Detoxifying enzyme (GST and AChE) activities in third instar larvae of *Plutella xylostella* fed with cabbage leaf discs treated with different concentrations of ethanol aqueous extract of leaf, bark, and seed oil of *T. sebifera* are presented (Figure 2). Results show that significant differences were observed between different concentrations after 48 h of treatment as compared to the untreated control. As the concentration increases the enzyme inhibition is decreased. In the GST activity, the higher concentration (2%) of ethanol aqueous extract of leaf, bark, and seed oil of *T. sebifera* significantly decreased the GST activity (Figure 2a) in *P. xylostella* (F_4,14_ = 978.78 to 1092; *p <* 0.0001) and was followed by other three concentrations (0.25 to 1%). However, the higher concentration of ethanol aqueous extract of the leaf at 2% resulted in lower GST activity, but significant differences were not found between leaf ethanol aqueous extract at 2 and 1%. Similarly, no significant differences in GST enzyme inhibition were observed between seed oil 0.5 and 1%. Similar results were also obtained in AChE (Figure 2b) feeding leaf discs treated with leaf, bark, and seed oil of *T. sebifera,* which significantly (F_4,14_ = 195.07 to 956.61; *p <* 0.0001) decreased the AChE activity in *P. xylostella*. Among the different concentrations studied, a higher concentration (2%) of ethanol aqueous extract of leaf, bark, and seed oil of *T. sebifera* significantly decreased the AChE activity in *P. xylostella* (*p <* 0.0001) and was followed by 1% as compared to the control. However, the enzyme inhibition/activity of leaf/bark ethanol aqueous extract 0.5% and seed oil 0.25% was increased as compared to the control.

## 3. Discussion

The qualitative analysis of metabolites in leaf and bark ethanol aqueous extract, antifeedant, repellent, and the joint action of leaf, bark ethanol aqueous extracts with seed oil, isolated compounds, and detoxification enzyme inhibition of *T. sebifera* against larvae of *P. xylostella* is discussed. Shikimic acid, xanthoxylin, quercetin, kaempferol, methyl gallate, and stigmasterol are the common metabolites identified in both leaf and bark ethanol aqueous extract. Apart from these, scopoletin, β-sitosterol, stigmasterol glycoside, and kaempferitrin along with one coumarin derivative compound and one glycosidic compound were identified from bark ethanol aqueous extract. In contrast, cinnamic acid, gallic acid, astragalin, and isoquercetin were identified in the leaf ethanol aqueous extract of *T. sebifera.*

In the current study, the leaf, bark extract and its binary mixtures, fractions, and seed oil of *T. sebifera* act as antifeedants against the larvae of *P. xylostella* and reduced the consumption of leaf. In the field conditions, the application of leaf/bark extract, its fractions, and seed oil on the plants/crops significantly reduce the damage by target pests and also showed promising efficacy against aphids [19]. In addition to the reduction in intake of food through their effects on chemoreceptors and taste receptors in the gustatory sensilla. During regular feeding, the taste receptors of insect pests are regularly exposed to complex mixtures of chemicals but no single molecule [26]. The results of the current study strongly suggest that the response of insects, whether feeding or non-feeding on the host plant, depends on chemical communication in the extracts/fractions/seed oil or the combinations/mixtures or chemical constituents detected by the gustatory sensilla. Initially, the starved larvae were not feeding but roving around the leaf discs. After settling on the leaf, the larvae started feeding on the leaf discs treated and untreated. However, larvae feed significantly more on untreated leaf discs as compared to treated ones. The feeding of larvae on cabbage leaves may be due to the presence of glucosinolates, which act as the feeding stimulant [27,28] and these responses are arbitrated by sensory receptors present in the mouthparts [29]. It is also reported that the chemicals that inhibit the feeding by plant-feeding insects may be the fundamental part of the plant defense that shows some measure of resistance to insect attack [30].

In the present study, bark ethanol aqueous extract reported higher feeding deterrence (DC_50_ = 2420.83–3678.02 mg/L as compared to leaf (DC_50_ = 3624.8–3944.50 mg/L) and seed oil (DC_50_ = 4019.85–9079.59 mg/L). Present results agree with the earlier reports, where ethanol extract of *Inula salsoloides* [31] and ethanol leaf/bark extract of *Strychnusnux vomica* [32] showed promising feeding deterrence against *P. xylostella* but are not superior to the current study. The present results also confirmed previous researchers who reported that neem seed oil and azadirachtin act as antifeedants against *P. xylostella* [33]. Binary mixtures of seed oil with bark ethanol aqueous extracts with and without choice showed higher feeding deterrence (DC_50_ = 317.10–1053.05 mg/L) as compared to seed oil with leaf extract (DC_50_ = 383.28–1328.66 mg/L). However, binary mixtures of seed oil with leaf ethanol aqueous extract (1:1 ratios) with choice and seed oil with bark and leaf ethanol aqueous extract without choice showed synergistic interaction. Present findings agree that the ethanol extract of *Piper retrofractum* + *Acorus calamus* against *S. litura* [34] and piperine + β-asarone and piperine + β-asarone against *S. litura* and *Mythimna separata* [35] showed deterrence. In a similar finding, the mixture of fraction 1, stigmasterol, acacetin, 20-hydroxyecdysone, and luteolin from ethanol extract of *Ajuga nipponensis* also showed synergistic activity against *P. xylostella* [36]. The feeding deterrence of leaf and bark ethanol aqueous extract and its combinations/mixtures in the current study might be due to the presence of common metabolites, viz., shikimic acid, xanthoxylin, quercetin, kaempferol, methyl gallate, and stigmasterol. The other metabolites present in the bark ethanol aqueous extract are scopoletin, β-sitosterol, stigmasterol glycoside, kaempferitrin, and coumarin/glycosidic derivative compounds. Similarly, the deterrent activity of seed oil may be due to the presence of saturated (6.73%) and unsaturated fatty acids (26.33%) as per earlier reports [19].

In the current investigation, leaf fractions showed higher deterrence as compared to bark. Among leaf fractions, the water and *n*-butanol fractions showed higher deterrence (DC_50_ = 219.88–235.75 mg/L) by the no choice method as compared to the ethyl acetate and *n*-hexane fractions. Similarly, the ethyl acetate and *n*-butanol fractions of bark showed higher deterrence (DC_50_ = 318.37–411.09 mg/L) as compared to the water and *n*-hexane fractions. The present study confirms the feeding deterrence of the ethyl acetate fraction of *Pergularia daemia* against *Helicoverpa armigera* and *S. litura* [37] and the butanol fraction of *Catunaregam spinosa* against *Pieris rapae* and *P. xylostella* [38]. The deterrence of the fractions against *P. xylostella* in the current study might be due to the presence of metabolites of hexadecanoic acid, galaxolide, ethyl phthalate, octadecanoic acid, ethyl ester, and 1-octadecene in the *n*-hexane fraction of leaf and bark [19]. The ethanol leaf extract (RC_50_ = 575.74 mg/L) showed more repellence against *P. xylostella* as compared to bark and seed oil. Among fractions, the *n*-butanol fraction of bark and water fraction of leaf (RC_50_ = 414.61–540.05 mg/L) showed promising repellence as compared to the *n*-hexane and ethyl acetate fractions (RC_50_ = 557.49–638.63 mg/L). Present results confirmed with the ethanol extract from fresh and dried de-oiled cake rhizomes [39] and seed oil of *Melia azedarach* showed comparatively low repellence against *S. littoralis* [40]. The repellent activity in the current studies also may be due to the presence of metabolites and fatty acids. Among isolated compounds, gallic acid showed higher feeding deterrence (63.80–67.48%) with and without choice as compared to kaempferol, quercetin, and shikimic acid (55.45–66.15%). Present results confirmed the previous study, where gallic acid isolated from *Alchornea glandulosa* leaf extract showed promising deterrence against neonate larvae of *S. frugiferda* [41].

The feeding inhibition of larvae may be either due to the masking of stimulant compounds in the leaf/bark, which provides neural inputs to the brain for initiation of feeding, or due to the presence of deterrent compounds in the plant extracts/fractions. In the current study, leaf and bark ethanol aqueous extract of *T. sebifera* showed 89.4 to100% inhibition of larval growth as compared to chloroform and petroleum ether extracts of *Pedicularis spicata* (72–74% inhibition) against *P. xylostella* [42]. Gallic acid, kaempferol, and quercetin isolated in the current study also showed higher feeding deterrence (64.6–67.4%) and were confirmed with triterpenoid saponins of *Clematis aethusifolia* [43], and eurycomanone from *Eurycoma longifolia* [44] inhibited the larval growth of *P. xylostella*. The *n*-butanol and water fractions of leaf and bark in this study showed growth inhibition against *P. xylostella* as compared to butanol and hexane fractions of *Gloriosa superba* seed extract against *S. litura* [45]. In another report, fraxinellone (20 mg mL^−1^) from *Dictamnus dasycarpus* reported an 85.4% reduction in the larval development of *Mythimna separata* [46]. The percent growth inhibition of leaf (68.65–100%) and bark fractions (79.92–99.68%) at 1000 ppm in the present study is higher than the previous report [45].

Insect pests developed resistance to broad-spectrum insecticides and few commonly used insecticides are effective against insect and mite pests to date. The insecticidal activities of botanical and synthetic molecules are dose-dependent [46,47]. Generally, insects utilize detoxifying enzymes to metabolize xenobiotics [48,49]. However, enzymes are also induced by botanical and synthetic pesticides, which plays important role in the development of resistance in insects [50,51].

GST is involved in the detoxification of different groups of insecticides [52,53]. The metabolites/compounds present in EOs and their mixtures, plant extracts, and seed oil also inhibit GST activity [54,55,56,57]. AChE influences the insect nervous system and hydrolyzes the acetylcholine neurotransmitter and terminates the nerve impulse in the synaptic cleft [58]. The inhibition of enzymes leads to the death of insects due to the over-accumulation of acetylcholine. GST is a major detoxifying enzyme, which metabolizes secondary metabolites present in the plants through catalysis of the conjugation of electrophile molecules and hydrolysis of ester bonds. However, the variation in the level of detoxifying enzyme activity may be due to environmental stress [59,60,61]. Higher concentrations of ethanol leaf/bark aqueous extract and the seed oil of *T. sebifera* in the present studies inhibited GST and AChE in *P. xylostella*. The current results confirmed the earlier study, where ginsenosides from leaves of *Panax ginseng* showed an inhibitory effect on GST and AChE in *P. xylostella* [62]. Essential oil of *Eucalyptus globulus* and *Allium sativum* decreased the AChE in *Ephestia kuehniella* [61]. All the concentrations of leaf and bark extract against larvae of *P. xylostella* in the current study showed an inhibitory effect on GST and AChE except leaf/bark extract 0.5% and seed oil 0.25%. The present results confirmed our previous studies, where the leaf, bark extract, and seed oil of *T. sebifera* showed significant inhibition of GST and AChE in *A. craccivora* [19]. The leaf extracts, seed oil, and their binary mixtures of *T. sebifera* showed excellent and promising antifeedant and synergistic activity against larvae of *P. xylostella.* Therefore, the extract, seed oil, and its combinations may be recommended as antifeedants under field conditions to reduce crop damage based on their persistence (safe waiting period), residue, phytotoxicity, safety to natural enemies of pests, and economics.

## 4. Material and Methods

### 4.1. Plant Material

The leaf, bark, and seeds of *T. sebifera* were collected in the farmer’s field (4000 sq. m area) at Chandpur (1466 m ASL), Tehsil Palampur (32°06′05″ N, 76°34′10″ E), District Kangra, Himachal Pradesh. The plant material was authenticated by Taxonomist, Division of Environmental Technology, CSIR-IHBT, Palampur, India. The voucher specimens (PLP 18563) were deposited in the herbarium. The plant material was dried under shade for 15 days and then used for the preparation of extract/fractions and seed oil for investigation.

### 4.2. Preparation of Leaf, Bark Extracts, and Fractions

Preparation of leaf, bark extracts, fractions, and extraction of seed oil was undertaken as per our earlier studies [19]. In this study, approximately 1 kg of dried powder of leaf and bark was macerated (5 L × 3 times) at room temperature using concentrations of 80% ethanol: water for 12 h. Filtered samples were then evaporated in a Rotavapor (Buchi, R-2010) at 45 °C under low pressure and the solvent was removed. Further, the dried ethanol aqueous extracts of the leaf and bark were sequentially fractionated using different solvents based on their polarity, i.e., *n*-hexane, ethyl acetate, *n*-butanol, and water. The compounds were isolated from the ethyl acetate and *n*-butanol fractions using the column chromatography method.

The seeds were dried for 15 days and the seed coat was taken out. The fat layer was removed using hot water treatment until the appearance of black kernels. The kernels were dried (1 kg) and macerated in *n*-hexane solvent (3 L × 4 times) at room temperature. The eluted solvent was evaporated under Rotavapor (Buchi, R-2010) at 45 °C under low pressure, which yielded 1 L of seed oil. The leaf and bark ethanol aqueous extracts, fractions, and seed oil were kept at less than 4 °C until further analysis.

### 4.3. Ultra-High-Performance Liquid Chromatography-Quadrupole Time of Flight-Ion Mobility Mass Spectrometry (UHPLC-QTOF-IMS) of Leaf and Bark Ethanol Aqueous Extract of T. sebifera

The dried extracts of leaf and bark were prepared for LC-MS by dissolving them in HPLC grade methanol to get a 10 mg mL^−1^ concentration and filtered using 0.25-micron syringe filters. The samples were injected into a high-resolution 6560 Ion Mobility QTOF LC-MS (Agilent Technologies, Santa Clara, CA, USA.). The column used for the metabolite separation was Eclipse Plus C18 RRHD (2.1 mm × 150 mm, 1.8 μm). The mobile phase comprised solvent A (0.1% formic acid in water) and solvent B (0.05% formic acid in ACN). The injection volume was 2 μL. A pre-published method was used to separate metabolites [63]. The metabolites were identified from total ion chromatograms (TICs) based on their *m*/*z* ratio, retention time (RT), and mass fragmentations (MS/MS). Data were processed using Mass Hunter Qualitative Analysis software (v. B.06.00, Agilent Technology).

### 4.4. Antifeedant Activities of Leaf, Bark Extracts, Fractions, Seed Oil, and Isolated Compounds of T. sebifera against P. xylostella

Antifeedant/feeding deterrence was tested using the choice [54] and no choice leaf disc assay [62]. Briefly, five concentrations of *ethanol aqueous* extract (625–10,000 mg/L), fractions (62.5–2000 mg/L), seed oil (625–10,000 mg/L), and isolated compounds (4000 mg/L) were prepared by serial dilution technique. Cabbage (*Brassica oleracea* L. var. *capitata* L.) leaf discs (4 cm diameter) were dipped in extracts, fractions, and seed oil emulsions for 10 s and air dried at room temperature for 1 h. The treated and control leaf discs were alternately placed in a Petri dish (90 mm diameter) lined with filter paper. A pair of third instars larvae of *P. xylostella* was starved for four hours and transferred to the center of the leaf discs to differentiate between the treated and the control leaf discs. The length between the two sets of opposite discs was about 1.5 cm. For control, leaf discs were dipped in distilled water containing 0.05 percent Tritone. Similarly, for the no choice method, the same methodology was followed as discussed above but the larvae were released on treated leaf discs only and then Petri dishes were kept under controlled conditions at 25 ± 2 °C temperature, 60 ± 5% relative humidity, and a photoperiod of 16 h light and 8 h dark. There were five treatments and each was replicated ten times. Indo Neem (Azadirachtin 0.15% EC) at the recommended dose (5 mL/L for the control of larvae of *P. xylostella* in the field conditions was used as a positive control for comparison. Data on the leaf area consumed by the larvae were measured before and 48 h after treatment using WinDIAS Image Analysis System (Delta-T Devices Ltd., Cambridge, UK). The feeding deterrence index (FDI) was calculated using the formula [64]. FDI = (C − T/C + T) × 100, where C is the average leaf area consumed in the control leaf disc and T is the average leaf area consumed in the treated leaf disc.

### 4.5. Growth Inhibition Activity

The methodology for the growth inhibition activity of *P. xylostella* is the same as discussed in the antifeedant activity assay. In this assay, larvae used for different concentrations were individually weighed before and 48 h after treatment. The growth inhibition rate (GIR) was calculated as per the formula used [65]. GIR = (C − T/C + T) × 100, where C is the average weight of the larvae in control and T is the average weight of the larvae in treatment.

### 4.6. Joint Action Studies

The methodology for joint action studies is the same as discussed in the antifeedant activity. In this study, five different concentrations (62.5–1000 mg/L) of the binary mixtures (1:1 ratios) of seed oil with leaf and bark ethanol aqueous extract of *T. sebifera* were prepared and evaluated against larvae of *P. xylostella*. The observations were recorded 72 and 96 h after treatment. The fractional effect indices (FEI) were calculated to study the joint action of binary mixtures/combinations. FEI = fractional effect of A + fractional effect of B, in which fractional effect of A = DC_50_ of mixture/DC_50_ of A and fractional effect of B = DC_50_ mixture/DC_50_ of B [66]. The FEIs were interpreted based on classifications [67] as being synergistic if FEI < 0.5, additive if FEI ≥ 0.5 and ≤1.0, indifferent if FEI > 1.0 and ≤4.0, or antagonistic if FEI > 4.0.

### 4.7. Repellent Activity 

The repellent activity of leaf, bark ethanol aqueous extracts, its fractions, and seed oil was studied against third instar larvae of *P. xylostella* by the choice test as per the methodology adopted [62]. Briefly, five concentrations of leaf, bark ethanol aqueous extracts, its fractions, and seed oil (625–10,000 mg/L) were prepared. Cabbage leaf discs (4 cm diameter) were dipped in the different concentrations for 10 s; shade dried and kept on a drawing sheet (72 cm × 56 cm) circularly at an equal distance for both treated and untreated leaf discs alternatively. About 25 third instar larvae were released at the center and allowed to settle on their choice in the leaf discs for 15 min. The observations on the number of larvae settled on treated and untreated leaf discs were recorded. The percent repellency (PR) was calculated by using the formula. PR = (C/T) × 100. Where C is the number of larvae settled in untreated leaf discs and T is the number of larvae settled in treated leaf discs.

### 4.8. Detoxification Enzyme Inhibition Activities of Leaf, Bark Extracts, and Seed Oil of T. sebifera against P. xylostella

Detoxification enzyme inhibition activities of leaf, bark extracts, and seed oil of *T. sebifera* against *P. xylostella* were performed as per the method followed [68]. Four different concentrations of leaf, bark extract, and seed oil (0.25, 0.5, 1, and 2%) were selected for enzyme activity assay according to the antifeedant assay described above. The larvae of each test concentration after 48 h of treatment were transferred into a centrifuge tube and were mixed with phosphate buffer (pH 7.4). The weight of larvae (mg): the volume of buffer (mL) was kept in a ratio of 1:9. These larvae were then homogenized with a homogenizer (Tarsons Micro Pestle). The homogenate was transferred immediately under ice bath conditions and then centrifuged at 15,000 rpm at 4 °C for 30 min. After that, the supernatant was taken into a new centrifuge tube for the test of enzyme activity. The total protein concentration was determined using the Bradford assay for all the samples before proceeding with the enzyme activity. For the determination of glutathione S-transferase (GST) and acetylcholine esterase (AChE), the assay kits for GST and AChE were procured from Cayman Chemicals (USA) and Abcam (UK), respectively.

### 4.9. Statistical Analysis 

The data on feeding deterrence, growth inhibition, synergistic, repellent activity, and their regression parameters were determined by Probit [69] using SPSS statistical software, version 16. The percent feeding deterrence, inhibition, repellency data of extracts, fractions, seed oil, compounds, and enzyme assay data were also analyzed by one-way analysis of variance (ANOVA) and means were compared by Tukey’s post hoc test [70]. 

## 5. Conclusions

In the present study, feeding deterrence, repellent, synergistic, and detoxification enzyme inhibition activities were studied. In a feeding deterrent assay, bark extract showed higher feeding deterrence and was followed by leaf extract. A combination of seed oil with bark ethanol aqueous extract was found to have more deterrence and was followed by seed oil with leaf ethanol aqueous extract. Among fractions, the ethyl acetate leaf fraction had more deterrence and was followed by *n*-butanol by without choice. Among compounds, gallic acid showed higher deterrence, and was at par with kaempferol and quercetin, than shikimic acid with choice. In the repellent assay, the *n*-butanol fraction of bark and the water fraction of leaf were more repellent, and were followed by the *n*-hexane and *n*-butanol fractions of leaf, than other fractions. In joint action studies, seed oil with leaf extract and bark extract at a 1:1 ratio showed promising deterrence and synergistic interaction against *P. xylostella*. A higher concentration of ethanol aqueous extract of leaf, bark, and seed oil of *T. sebifera* at 2% significantly decreased the GST and AChE activity in *P. xylostella.*

## Figures and Tables

**Figure 1 molecules-27-06239-f001:**
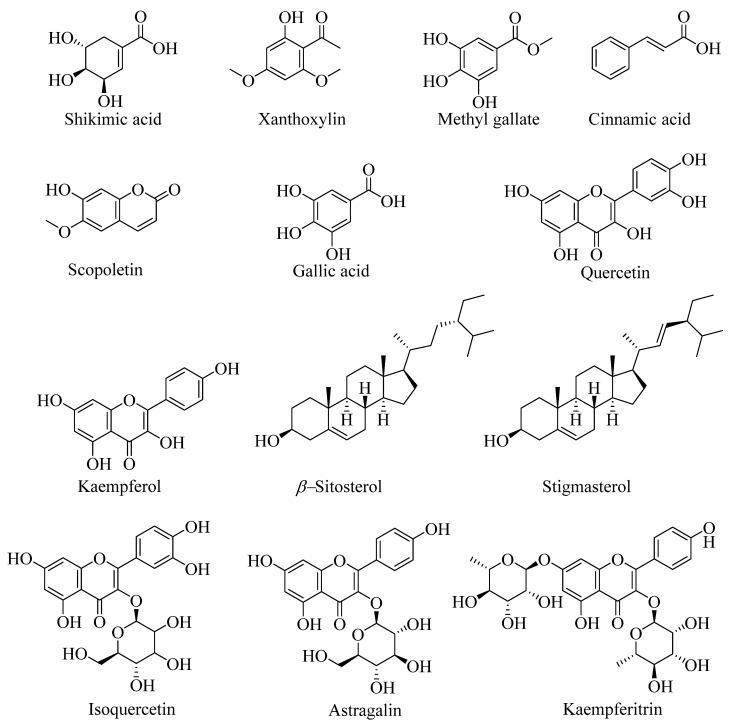
Structures of identified metabolites from leaf and bark ethanol aqueous extracts of *Triadica sebifera*.

**Figure 2 molecules-27-06239-f002:**
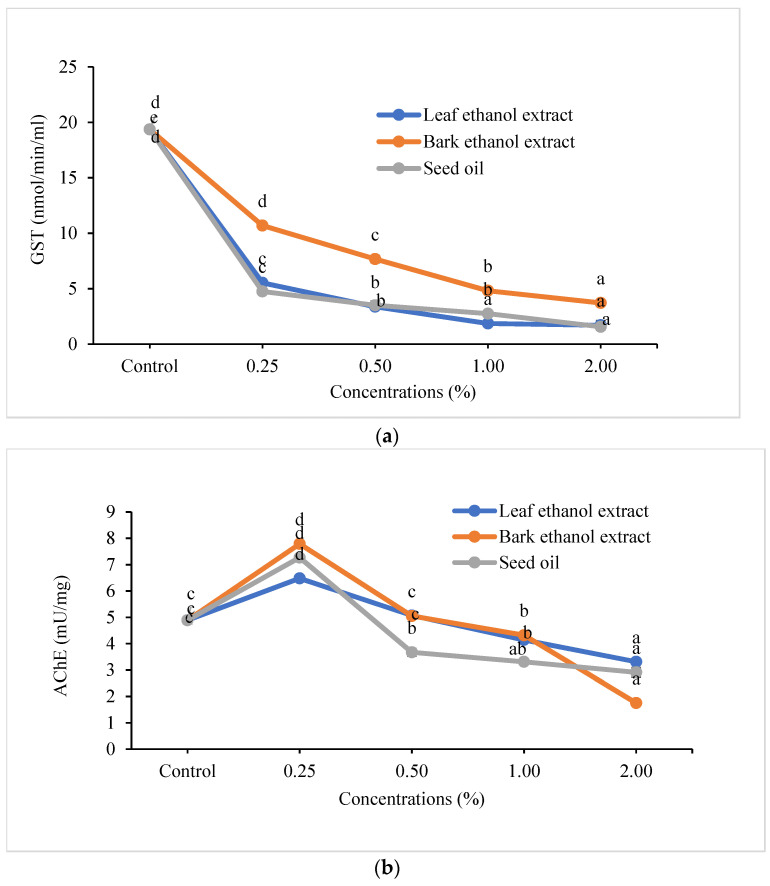
(**a**) Glutathione S-transferase enzyme (GST) inhibition in *Plutella xylostella* treated with leaf, bark ethanol aqueous extracts, and seed oil of *Triadica sebifera*; Mean of three replications (*n* = 3); The same letters in the error bars (Mean ± SE) within the figure are not statistically different by Tukey’s HSD (*p* ≥ 0.05). (**b**). Acetylcholine esterase enzyme (AChE) inhibition in *Plutella xylostella* treated with leaf, bark ethanol aqueous extracts, and seed oil of *Triadica sebifera*; Mean of three replications (*n* = 3); The same letters in the error bars (Mean ± SE) within the figure are not statistically different by Tukey’s HSD (*p* ≥ 0.05).

**Table 1 molecules-27-06239-t001:** Identified metabolites in leaf and bark ethanol aqueous extract of *Triadica sebifera* using UHPLC-QTOF-IMS.

RT	Identified Compounds	Chemical Formula	Observed Mass (M+H)^+^/(M+Na)^+^	Mass Fragments	Leaf Extract *	Bark Extract *	References
4.155	Shikimic acid (**1**)	C_7_H_10_O_5_	175.15 (M+H)^+^	175 (M+H)^+^, 174 [C_7_H_10_O_5_ (M)]^+^, 130 [C_6_H_10_O_3_ (M-COOH)]^+^	+	+	[20,21]
4.155	Xanthoxylin (**2**)	C_10_H_12_O_4_	197.13 (M+H)^+^	197 (M+H)^+^, 196 [C_10_H_12_O_4_ (M)]^+^, 180 [C_10_H_12_O_3_ (M-OH)]^+^	+	+	[22]
4.705	Quercetin (**3**)	C_15_H_10_O_7_	303.10 (M+H)^+^	303 (M+H)^+^, 197 [C_9_H_8_O_5_ (M+H-C_6_H_6_O_2_)]^+^	+	+	[20,21]
4.889	Kaempferol (**4**)	C_15_H_10_O_6_	287.11 (M+H)^+^	287 (M+H)^+^, 182 [C_9_H_8_O_4_ (M+2H-C_6_H_6_O_2_)]^+^	+	+	[20,21]
5.739	Methyl gallate (**5**)	C_8_H_8_O_5_	207.02 (M+Na)^+^, 185.04 (M+H)^+^	207.02 (M+Na)^+^, 185.04 (M+H)^+^	+	+	[21,23]
6.538	Coumarin derivative compound (**6**)	C_24_H_30_O_4_	383.13 (M+H)^+^	383 (M+H)^+^, 163 (C_9_H_6_O_3_+H)^+^	-	+	
7.455	Glycosidic compound (**7**)	-	503.16 (M+H)^+^	503 (M+H)^+^, 341 ((M+H-glu)^+^)	-	+	
7.557	Cinnamic acid (**8**)	C_9_H_8_O_2_	149.11 (M+H)^+^	149(M+H)^+^, 148 (M)^+^	+	-	[21,24]
7.639	Scopoletin (**9**)	C_10_H_8_O_4_	193.05 (M+H)^+^	193 (M+H)^+^, 163 [C_9_H_6_O_3_ (M+H-OCH_3_)]^+^	-	+	[21,22]
7.682	Gallic acid (**10**)	C_7_H_6_O_5_	341.11 (2M+H)^+^, 171.09 (M+H)^+^	171.09 (M+H)^+^, 127 [C_6_H_6_O_3_ (M+H-COOH)]^+^	+	-	[20,21]
8.187	β-sitosterol (**11**)	C_29_H_50_O	415.15 (M+H)^+^	415 (M+H)^+^, 398 [C_29_H_50_ (M-OH)]^+^	-	+	[21,25]
8.367	Stigmasterol (**12**)	C_29_H_48_O	413.20 (M+H)^+^	413.20 (M+H)^+^, 397 [C_29_H_48_ (M+H-OH)]^+^	+	+	[21,25]
12.647	Stigmasterol glycoside (**13**)	C_35_H_58_O_6_	575.20 (M+H)^+^	575 (M+H)^+^, 413 [C_29_H_48_O (M+H-glu)]^+^	-	+	
12.759	Astragalin (**14**)	C_21_H_20_O_11_	449.17 (M+H)^+^	449 (M+H)^+^, 287 [C_15_H_10_O_6_ (M+H-glu)]^+^	+	-	[20,21]
13.187	Isoquercetin (**15**)	C_21_H_20_O_12_	487.07 (M+Na)^+^, 465.09 (M+H)^+^	487 (M+Na)^+^, 465 (M+H)^+^, 303 [C_15_H_10_O_7_ (M+H-glu)]^+^	+	-	[20,21]
15.474	Kaempferitrin (**16**)	C_27_H_30_O_14_	579 (M+H)^+^	579 (M+H)^+^, 433 [C_21_H_20_O_10_ (M+H-rha)]^+^, 287 [C_15_H_10_O_6_ (M+H-rha-rha)]^+^	-	+	[21]
18.794	Unidentified (**17**)	-	325.22		+	+	

* + means present, - means absent.

**Table 2 molecules-27-06239-t002:** Feeding deterrence of leaf, bark ethanol aqueous extracts, and seed oil of *Triadica*
*sebifera* against *Plutella xylostella*.

Extracts/Oils	DC_50_ (mg/L)	Confidence Limits (mg/L)	Slope ± SE	Chi Square	*p* Value
**With choice**					
Leaf extract	3624.80	3001.45–4477.46	1.45 ± 0.15	4.56	0.21
Bark extract	2420.83	1843.70–3159.31	1.01 ± 0.14	5.28	0.15
Seed oil	9079.59	6076.20–17,945.33	0.86 ± 0.14	1.17	0.76
**Without choice**					
Leaf extract	3944.50	3352.00–4733.27	1.76 ± 0.16	1.70	0.64
Bark extract	3678.02	3022.92–4595.49	1.39 ± 0.15	5.04	0.17
Seed oil	4019.85	3277.78–5106.26	1.35 ± 0.15	4.61	0.20
**Binary mixtures**					
**With choice**					
Seed oil + Leaf extract	1328.66	935.19–2202.52	0.86 ± 0.11	3.94	0.41
Seed oil + Bark extract	1053.05	704.34–2069.16	0.93 ± 0.15	2.03	0.57
**Without choice**					
Seed oil + Leaf extract	383.28	318.71–473.16	1.51 ± 0.15	4.99	0.17
Seed oil + Bark extract	317.10	272.38–371.91	1.86 ± 0.16	4.68	0.20
Indo-Neem (Choice)	2024.58	1016.54–9451.10	0.38 ± 0.13	0.32	0.96
Indo-Neem (No choice)	2873.99	2219.32–4079.19	1.15 ± 0.15	0.63	0.89

**Table 3 molecules-27-06239-t003:** Feeding deterrence of leaf and bark fractions of *T**riadica*
*sebifera* against *Plutella xylostella*.

Leaf Fractions	DC_50_ (mg/L)	Confidence Limits (mg/L)	Slope ± SE	Chi Square	*p* Value
**With choice**					
*n-*Hexane	755.51	543.88–1247.65	0.99 ± 0.14	1.53	0.67
Ethyl acetate	265.16	218.95–322.17	1.43 ± 0.15	4.73	0.19
*n-*Butanol	1063.14	742.51–1785.27	0.77 ± 0.11	2.23	0.69
Water	1504.53	960.59–3110.55	0.68 ± 0.11	1.42	0.84
**Without choice**					
*n-*Hexane	577.71	432.70–868.70	1.02 ± 0.14	2.67	0.44
Ethyl acetate	283.68	235.77–343.68	1.47 ± 0.15	2.31	0.51
*n-*Butanol	235.75	191.66–288.37	1.35 ± 0.15	4.31	0.23
Water	219.88	173.71–274.58	1.20 ± 0.14	4.66	0.20
**Bark fractions**					
**With choice**					
*n-*Hexane	455.41	361.78–606.92	1.21 ± 0.15	4.97	0.17
Ethyl acetate	727.68	505.14–1311.72	0.86 ± 0.14	0.15	0.97
*n-*Butanol	2189.43	1344.16–4921.91	0.73 ± 0.11	3.67	0.45
Water	1447.12	1040.79–2305.51	0.96 ± 0.11	3.32	0.51
**Without choice**					
*n-*Hexane	573.02	435.01–839.05	1.07 ± 0.14	1.53	0.68
Ethyl acetate	318.37	270.53–378.44	1.71 ± 0.16	3.98	0.26
*n-*Butanol	411.09	356.19–481.11	2.07 ± 0.17	4.89	0.18
Water	488.20	414.43–590.65	1.84 ± 0.17	5.28	0.15
Indo-Neem (Choice)	2024.58	1016.54–9451.10	0.38 ± 0.13	0.32	0.96
Indo-Neem (No choice)	2873.99	2219.32–4079.19	1.15 ± 0.15	0.63	0.89

**Table 4 molecules-27-06239-t004:** Feeding deterrence of isolated compounds (4000 mg/L) of *Triadica sebifera* against *P. xylostella*.

Compounds	Percent Feeding Deterrence Index (±SE) after 48 h of Treatment *		Percent Growth Inhibition (±SE)
	With Choice	Without Choice	Without Choice
Kaempferol-3-*O*-glucoside	66.15 ± 2.10 a	55.75 ± 1.25 b	45.08 ± 2.50 c
Quercetin-3-*O*-glucoside	64.63 ± 1.96 a	57.63 ± 1.54 ab	57.34 ± 2.29 b
Gallic acid	67.48 ± 1.99 a	63.8 ± 2.39 a	59.02 ± 1.57 b
Shikimic acid	56.50 ± 1.78 b	55.45 ± 1.48 b	57.52 ± 1.58 b
Indo-Neem (5 mL L^−1^)	54.97 ± 2.25 b	58.02 ± 2.56 a	65.68 ± 1.60 a
F_4,49_	8.20; *p* < 0.0001	3.13; *p* < 0.024	14.56; *p* < 0.0001

* Mean of three replications; Mean followed by the same letters within a column are not statistically different by Tukey’s HSD (*p* ≥ 0.05).

**Table 5 molecules-27-06239-t005:** Repellent activity of leaf/bark extracts, fractions, and seed oil of *Triadica*
*sebifera* against *Plutella xylostella*.

Leaf/Bark Extracts/Oil	RC_50_ (mg/L)	Confidence Limit (mg/L)	Slope ± SE	Chi Square	*p* Value
Leaf extract	575.74	249.81–910.78	0.80 ± 0.14	0.74	0.86
Bark extract	628.02	288.20–972.01	0.81 ± 0.14	0.60	0.90
Seed oil	630.87	382.65–874.28	1.16 ± 0.16	3.93	0.27
**Fractions (Leaf)**					
*n-*Hexane	557.49	274.05–845.48	0.92 ± 0.15	0.52	0.91
Ethyl acetate	638.63	230.69–1059.34	0.67 ± 0.14	1.81	0.61
*n-*Butanol	565.18	223.21–920.87	0.75 ± 0.14	1.54	0.67
Water	540.05	247.03–840.55	0.87 ± 0.15	0.92	0.82
**Fractions (Bark)**					
*n-*Hexane	629.58	388.77–866.07	1.12 ± 0.16	0.77	0.86
Ethyl acetate	773.76	494.19–1047.82	1.13 ± 0.15	2.24	0.52
*n-*Butanol	414.61	82.32–807.53	0.59 ± 0.14	1.03	0.79
Water	834.48	445.86–1219.92	0.83 ± 0.14	0.46	0.93

**Table 6 molecules-27-06239-t006:** Joint action activities of seed oil with leaf and bark extracts of *Triadica*
*sebifera* against *Plutella xylostella*.

**Binary Mixtures**	**% FDI (mg/L)**	**FEI**	**Interaction Type**
**With choice**			
Seed oil + leaf extract (1:1)	49.57	1.634	Indifferent
Seed oil + bark extract (1:1)	51.91	1.493	Indifferent
**Without choice**			
Seed oil + leaf extract (1:1)	79.62	2.211	Indifferent
Seed oil + bark extract (1:1)	77.75	2.122	Indifferent
**Binary mixtures**	**DC_50_ (mg/L)**	**FEI**	**Interaction type**
**With choice**			
Seed oil + leaf extract (1:1)	1328.66	0.513	Synergistic
Seed oil + bark extract (1:1)	1053.05	0.726	Additive
**Without choice**			
Seed oil + leaf extract (1:1)	383.28	0.193	Synergistic
Seed oil + bark extract (1:1)	317.10	0.167	Synergistic
**Binary mixtures**	**% GIR (mg/L)**	**FEI**	**Interaction type**
**Without choice**			
Seed oil + leaf extract (1:1)	99.87	2.329	Indifferent
Seed oil + bark extract (1:1)	98.77	2.090	Indifferent

FDI—Feeding deterrence index; FEI—Fractional effect indices; GIR—Growth inhibition rate.

## Data Availability

Not applicable.

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
