# Peer review of "Characterization of *Triadica sebifera* (L.) Small Extracts, Antifeedant Activities of Extracts, Fractions, Seed Oil and Isolated Compounds against *Plutella xylostella* (L.) and Their Effect on Detoxification Enzymes"

_molecules, 2022, doi:10.3390/molecules27196239_

Round 1
Reviewer 1 Report
The manuscript describes the evaluation of the biological properties of Triadica sebifera against Plutella xylostella. Although the authors have observed some interesting results of biological activity, the chemical analysis is not properly described. The text needs in-depth revision and the argumentation and discussion contains conceptual errors.
Specific points to be reviewed are highlighted as follow.
Abstract:
In the abstract, the sentence that should highlight the reason for choosing the plant species should be accompanied by a logical argument about previous works that indicate that the species already has potential against insects, or even the availability of plant material to be considered as a potential insecticide. However, the authors only mention that “Insecticidal activities of Triadica sebifera (L.) was not documented against P. xylostella till date.”. I recommend reformulating the abstract.
Still in the abstract and title, when mentioning synergistic activity, it is always necessary to clarify what type of synergism the author is considering, and which species, extracts, molecules, etc., are involved in the synergistic action.
Results
Line 62: “A total of sixteen peaks were identified in leaf and bark ethanol extracts of T. sebifera, while one major peak present in both samples remains unidentified.” The analytical tool used allows the ANNOTATION of compounds. Only with co-injection with authentic or commercial standards the term “identification” is appropriate, otherwise terms such as “compound annotation” should be employed. There is no mention of injections of standards in p.14 line 332.
Several times throughout the text “ethanol extract” (line 65) or “aqueous ethanol extract” (line 75) is mentioned. Clarify which extract is being used.
Linha 77: “…with molecular ion peak at m/z 175.15 (M+H)+ ”. Molecular ion refers exclusively to molecular mass in ionic forms, such as M+ or M-. Therefore, M+H is not a molecular ion, it is a protonated molecule with m/z equivalent to M+H.
The annotation\identification of flavonoids, whether genins or glycosides, is well established and takes into account the specific fragmentation such as via retro-Dials Alder and losses of sugar moieties. The annotation of the peaks needs to be better discussed considering all information obtained by the MS data, in order to characterize these already well-established fragments, since these types of fragments show important characteristics of the molecules, such as the arrangement of substituents in the rings of flavonoid skeleton.
Line 89: An ion with m/z corresponding to [M+H]+ can be considered a fragment? Is it possible an ion [M+H]+ and/or aducts [M+Na]+ to be named a fragment, such as in the phrase “Similarly, methyl gallate (5) was further identified from its mass fragments observed at m/z 207.02 (M+Na)+ , 185.04 (M+H)+.”?
Table 1. : What are the molecular formulas of compounds 6 and 17? Even without being annotated, the authors must add the possible molecular formulas, with the smallest errors, in the table, as these are important compounds in the chemical profile. What is the molecular formula and what is the peak number in the chemical profile of ”one major peak present in both samples remains unidentified.” (p.2 line 66)
p.6 line 37: “significantly (F4, 49=5.42 to 24.19; p<0.001)” What is the meaning of F4.49? How was it prepared?
Discussion:
First paragraph of the discussion repeats the same information as the results, and does not add anything new to the text.
p. 12 line 218: 19]. “In addition to the reduction in intake of through their effects on chemoreceptors and taste receptors in the gustatory sensilla.” The authors begin the discussion with a statement, for which the experiments were not designed. I suggest to discuss the findings as an evidence, and to show data from the literature that support this observation, removing that aforementioned sentence and starting the discussion with “During regular feeding, the taste receptors of insect pests are regularly exposed to complex mixtures of chemicals but no single molecule [26]. The results of the current study strongly suggest that the response of insects whether feeding or non-feeding on the host plant depends on chemical communication in the extracts/fractions/seed oil or its combinations/mixtures or chemical constituents detected by the gustatory sensilla.”
Line 244: “Present findings agree with that of ethanol extract of Piper retrofractum + Acorus calamus against S. litura [34] and piperine + β-asraone and piperine + β-asraone against S. litura and Mythimna separata [35].” Why do the authors mention different combinations of compounds and different organisms? The results of the present work do not agree with previous research, because previous research deals with other combinations of compounds and other organisms. They're not even the same molecules, so using this example is meaningless. The same fact is repeated on p. 12 and 13 and lines 261, 271, 276, 310 and so on. β-asraone (asarone) is misspelled.
Line 315: It is mentioned that “Present results confirmed with our previous studies, where the leaf, bark extract and seed oil of T. sebifera showed significant inhibition of GST and AChE in A. craccivora [63].” However, the reference in question is “63. Dadwal, V.; Joshi, R.; Gupta, M. A multidimensional UHPLC-DAD-QTOF-IMS gradient approach for qualitative and quantita-601 tive investigation of citrus and malus fruit phenolic extracts and edibles. ACS Food Science & Technology 2021, 1, 2006”. This work has no mention of the participation of the authors of this manuscript
p. 13 line 319: “Therefore, the extract, seed oil and its combinations can be recommended as antifeedant under field conditions to reduce crop damage.” It is a serious mistake to recommend the use of extracts or bioactive combinations without having undergone appropriate toxicity and environmental risk tests. The use of substances, even from natural origin, cannot be recommended without proper toxicity assessment.
p. 14 line 329: Even with the mention of the previous article of the group, it is essential to report, even if briefly, the method of obtaining the extracts and substances discussed in the present work.
Reviewer 2 Report
The manuscript entitled "Characterization of Plant Extracts, Antifeedant, Synergistic Activities of Extracts, Fractions and Isolated Compounds from Triadica sebifera (L.) Small against Plutella xylostella (L.) and their Effect on Detoxification Enzymes" is exceptionally well-written and I was unable to find scientific faults. please replace the sentence Soviet Union with Russian Federation.
Reviewer 3 Report
The work placed in the manuscript is good, it is of great interest and it is very complete. The introduction is well written and shows a good knowledge of the topic. The work presents a good methodology, which is already known, is well explained and the results are good. The methods are well described and well explained, the comments and the comparisons of the values are good. The conclusions presented in the work are well described and understood correctly.
On page 13, line 312, the word should be corrected: globules, by globulus. In general, the text must be revised in the signs, some point may be missing.
